# Lifestyle, Type of Work, and Temporary Disability: An Incidence Study of the Working Population

**DOI:** 10.3390/ijerph192214932

**Published:** 2022-11-13

**Authors:** Rocío Jiménez-Mérida, Manuel Romero-Saldaña, Domingo de-Pedro-Jiménez, José Manuel Alcaide-Leyva, Vanesa Cantón-Habas, Carlos Álvarez-Fernández, Manuel Vaquero-Abellán

**Affiliations:** 1Departamento de Enfermería, Farmacología y Fisioterapia, Facultad de Medicina y Enfermería, Universidad de Córdoba, 14014 Córdoba, Spain; 2Grupo Asociado de Investigación GA16 Estilos de Vida, Tecnología y Salud, Instituto Maimónides de Investigación Biomédica de Córdoba (IMIBIC), Departamento de Enfermería, Farmacología y Fisioterapia, Facultad de Medicina y Enfermería, Universidad de Córdoba, 14014 Córdoba, Spain; 3Departamento de Enfermería, Facultad de Enfermería, Universidad de Cádiz, 14014 Córdoba, Spain; 4Departamento de Salud Laboral, Ayuntamiento de Córdob, 14008 Córdoba, Spain

**Keywords:** health promotion, Mediterranean diet, physical activity, sick leave, workplace

## Abstract

The objective of the study was to identify lifestyles associated with loss of health among workers. A retrospective longitudinal incidence study was carried out over a three-year period (2015, 2016, and 2017) among the working population. A total of 240 workers were analysed using information from occupational health assessments. The outcome variable was loss of health due to common illness or workplace injury, quantified by the number of days each episode lasted. Predictor variables were age, gender, type of work, tobacco use, alcohol consumption, physical activity (IPAQ), and adherence to the Mediterranean diet (AMD). An adjusted multiple linear regression was performed, determining the goodness of fit of the final model using the coefficient of determination adjusted *r*^2^. During the study, 104 men (58.8%) and 25 women (39.7%) suffered an episode of illness or workplace injury (*p* < 0.05). The overall incidence was 17.9% people/year 95% CI [15, 21.3]. 4.6% of the workers were sedentary or engaged in light physical activity, and 59.2% maintained an adequate AMD. Workers who engaged in high levels of physical activity had an average of 36.3 days of temporary disability compared to 64.4 days for workers with low-moderate levels of physical activity (*p* < 0.01).

## 1. Introduction

Chronic illness caused by unhealthy lifestyles accounts for a huge amount of health and social spending around the world [1,2]. It is also linked to high levels of employee absenteeism and sick leave, as well as reduced productivity and poorer professional performance [3]. For this reason, companies are increasingly opting to organise health promotion activities in the workplace to encourage healthy eating habits and increase adherence to physical activity (PA) among employees with the aim of improving their health and wellbeing, creating a better working environment, and enhancing productivity in the workplace [4]. The main health problems addressed through these activities and linked to higher levels of employee absenteeism are overweightness, obesity and sedentarism [5,6,7] along with other unhealthy habits such as alcohol consumption and tobacco use. 

Healthy eating habits include adherence to the Mediterranean diet (AMD) as a starting point for improving workers’ diets, which is based on eating seasonal fruit and vegetables, fish, olive oil as a primary source of fat, and pulses, among other foods [8]. This type of diet has been linked to a lower risk of heart disease [9] and metabolic syndrome [10], effective management of obesity and overweightness [11], and the prevention of some types of cancer [12,13,14].

Physical activity (PA) is another key component involved in improving public health and plays a central role in personal wellbeing. Activities such as aerobic exercise, yoga, pilates [15,16], breaks during working hours for people in sedentary roles [17], and outdoor exercise [18] have been shown to have a positive impact on reducing sedentarism and obesity among workers [19]. However, many companies find it difficult to fit this type of activity into the working day [20].

The cost of absenteeism, temporary disability (TD), and workplace illnesses pose a challenge for many workplaces and for society more broadly. Health promotion activities in the workplace are essential for companies that wish to improve their employees’ wellbeing [3,21]. Nevertheless, these interventions must be adapted to the specific needs of each company, the type of work, and the personal characteristics of employees. To achieve this, workers’ lifestyles and loss of health, as well as their impact on absenteeism, should be analysed before any intervention is carried out. 

The research hypothesis was to find out if those workers who performed moderate-high physical activity or high adherence to the Mediterranean diet presented less loss of health due to temporary disability.

This study aimed to identify lifestyles associated with loss of health among workers by analysing (1) the relationship between AMD and loss of health, and (2) the relationship between different levels of PA and loss of health among workers.

## 2. Material and Methods

### 2.1. Study Design

This was a retrospective longitudinal incidence study carried out over a three-year period (2015, 2016, and 2017).

### 2.2. Population and Sample

The reference population was constituted by the workers of Córdoba City Council (Spain), with an average population of 1500 workers.

The necessary simple size was calculated using EPIDAT 4.2. For an expected prevalence of 68% high AMD, with 95% certainty and 6% accuracy, the sample size was estimated to be 202 workers. Participants were selected through random sampling stratified by age and sex.

### 2.3. Eligibility Criteria

The participants were recruited from regular occupational health checkups that they carried out in the company.

Inclusion criteria. Workers who had attended an occupational health assessment during the study period and were working at the start of the study were included.

Exclusion criteria. Workers who had been at their current workplace for less than 3 years, workers who had undergone health assessments upon starting their jobs, and workers who were on sick leave due to cancer during the study period were excluded from the study.

### 2.4. Study Variables and Measurement

Outcome variable: loss of health (days), measured as the total number of days of sick leave taken by the worker due to TD, either due to common illness (CI) or workplace injury (WI). Information on TD was obtained from workers’ occupational health records.

Explanatory variables: age (years), sex (male, female), alcohol consumption (measured according to World Health Organization [22] and classifying the participants in the following categories: (i) none–low and (ii) moderate–high; tobacco use (non-smoker, smoker, ex-smoker); type of work: administrative professions (technical and administrative personnel), security professions (local police officers and firefighters), and trade professions (gardeners, builders, electricians, etc.). PA (METs) was measured using the short form version of the International Physical Activity Questionnaire (IPAQ) [23], which classifies workers as follows: sedentary–light PA (<600 METs per week), moderate PA (600–2999 METs per week), vigorous PA (≥3000 METs per week). AMD was measured using the modified Medas questionnaire [24], which classifies workers into low adherence (<9 points) and high adherence (≥9 points).

### 2.5. Ethical and Legal Aspects

The study was carried out in accordance with the ethical principles for medical research involving human subjects set out in the Declaration of Helsinki. All the workers were informed of the study objectives verbally and in writing, and written informed consent was obtained. This research was approved by the Andalusian Biomedical Research Committee (Spain) with document number 4427/295.

### 2.6. Data Analysis

Quantitative variables were expressed as means and standard deviations as well as median and interquartile range. Qualitative variables were expressed as absolute and relative frequencies (percentages). The assumption of normality was assessed using the Kolmogorov–Smirnov test with the Lilliefors correction (*n* > 50) and the Shapiro–Wilk test (*n* < 50). 

The assumption of homogeneity of variances was assessed using Levene’s test. To compare two independent samples, the Student’s *t*-test (parametric) or Mann–Whitney’s *U*-test (non-parametric) were used. To compare more than two samples, the ANOVA or Kruskal–Wallis tests were used, depending on the parametric nature of the samples. Percentages were compared using the chi-squared test. 

Incidence of TD was calculated and a multiple linear regression was performed, determining the goodness of fit of the final model using the coefficient of determination adjusted *r*^2^. Fulfilment of the conditions of collinearity (via analysis of the variance inflation factor), normality, and independence of residuals was validated using the normality and Durbin–Watson tests respectively.

All tests were conducted with a statistical significance threshold of alpha error < 5%. Confidence intervals (CIs) were calculated with 95% certainty.

## 3. Results

The study sample comprised 240 workers, 177 of whom were men (73.8%) and 63 of whom were women (26.2%). Their average age was 50.2 (SD = 7.9) 95% IC [49.2, 51.2]. 

Table 1 shows the characteristics of the study sample in comparison with the reference population. No significant differences were observed, with the exception of the type of work as administrative workers attending the most occupational health assessments during the study period. When the sample was disaggregated by type of work, the majority of the sample were administrative workers (45.4%), followed by police officers and firefighters (30%), and tradespeople (24.6%).

### 3.1. Temporary Disability

During the three-year study period, 129 workers experienced an episode of TD due to either CI or WI (Figure 1), resulting in an overall incidence of 17.9% people/year 95% CI [15, 21.3]. By gender, 104 were men, giving an incidence of 19.6% people/year 95% CI [16,23.7], and 25 were women, giving an incidence of 13.2% people/year 95% CI [8.6, 19.5] (*p* = 0.09).

The mean duration of TD (CI and/or WI) was 46.1 (99.3) days 95% CI [33.4, 58.7], with a range of 0–741 days. Table 2 shows the mean duration of TD in relation to several of the study variables. Significant differences were observed (*p* < 0.01) by type of work: the administrative workers had the lowest mean number of days lost due to TD (29.5) compared to 47.8 days among security workers and 74.5 days among tradespeople.

### 3.2. Adherence to the Mediterranean Diet and Physical Activity

As for AMD, the mean for the whole sample was 8.8 (1.8) 95% CI [8.6, 9.1]), with a mean of 8.9 (SD = 1.9) for men and 8.8 (SD = 1.7) for women (*p* = 0.68). A total of 142 workers (59.2%) had a high AMD (>9 points), with similar percentages for men (58.2%) and women (61.9%) (*p* = 0.61). By type of work, the administrative workers obtained an AMD of 50.5%, trade professions of 61%, and security workers of 70.8% (*p <* 0.05).

Finally, Table 3 shows the relationship between the components in the AMD questionnaire and the mean duration of TD. No significant association was found between any component of AMD and the duration of TD.

With regard to PA, 11 workers (4.6%) indicated that they engaged in light PA or were sedentary, 72 (30%) engaged in moderate PA, and 157 (65.4%) engaged in vigorous PA. Table 4 shows the results for the mean duration of TD and the level of PA among workers, classified into three different categories (sedentary–light, moderate, vigorous), two different categories (high and not high), and by MET quartiles.

Workers with a high level of PA had a lower mean duration of TD at 36.3 (69.4) days than workers without a high PA (sedentary, light or moderate), among whom the mean duration was 64.4 (138.1) days (*p* = 0.085). This was also demonstrated by comparing the mean duration of TD by PA quartiles: the workers belonging to Q2, with a PA between 2455 and 4000 METs per week, had the lowest mean duration (19.2 days), compared to 84 days in the group with the lowest PA (Q1), at fewer than 2455 METs per week.

Finally, a multiple linear regression was performed to ascertain the influence of AMD and PA on the duration of TD (outcome variable). This was adjusted for the main study variables (age, gender, tobacco use, alcohol consumption, and type of work). Table 5 shows how type of work and PA are the only two significant variables for the total duration of TD, while AMD is not associated with the length of sick leave due to common illness and/or workplace injury.

With regard to type of work, manual workers (trade professions) obtained a coefficient of 51.5 (*p* < 0.001) and local police officers and firefighters (security professions) obtained a coefficient of 24 (*p* = 0.11) compared to the reference category, which was made up of administrative workers.

Workers with a high PA (>3000 METs per week) obtained a coefficient of −35.4 in relation to the reference category, which comprised workers with a sedentary-light-moderate PA (*p* < 0.01).

## 4. Discussion

This longitudinal study investigated the impact of AMD and PA on workers’ health, expressed as loss of health in the form of episodes of TD.

A significant association was observed between high levels of PA and a shorter duration of TD, compared to workers with a sedentary-low-moderate level of PA, who took more days off due to TD. Similar results have been reported in other studies [25,26] corroborating the relationship between vigorous PA and reduced employee absenteeism. Lahti et al. [27] showed that, when exercising for the same length of time at different intensities, the risk of TD for the group exercising moderately was not reduced while the risk for the group exercising vigorously was. Other studies [25], however, found no relationship between low and moderate levels of PA and a reduction in employee absenteeism. In Herruzo et al. [28], 79.9% of workers engaged in sedentary–moderate PA while 16% engaged in vigorous PA. These results differ from the findings in our study, and it is possible that the type of work influenced the type of PA performed by workers.

It has been proven that engaging in vigorous PA three times a week can be effective in reducing TD [29]. However, it is not immediately clear why workers do not engage in vigorous PA. It is possible that workers do not engage in PA or do so with low–moderate intensity because their physical fitness does not permit them to perform vigorous exercise or simply because they do not wish to do so. Social determinants of health also have a huge impact on workers’ PA opportunities. Some authors highlight that PA intervention strategies may target/address social aspects such as economic level, environment, and access to health and social services [30,31]. Therefore, it is essential to consider those aspects and study vulnerable workers’ needs, designing safe and healthy spaces in the workplace [32], as well as strong and supportive policies that help overcome those barriers [33,34].

Another aspect to consider is workers’ evolution over the course of the study; when data collection began, they may have engaged in vigorous PA before ceasing to do so or shifting to moderate PA over time due to their age or health [29].

With regard to the Mediterranean diet, multiple studies have demonstrated that a high AMD serves as a protective factor against chronic degenerative diseases and improves personal wellbeing, making it an excellent way of maintaining good health [12,35,36]. Although high AMD has been shown to protect against certain diseases, the findings of this study showed no significant association between AMD and duration of TD. It is possible that no positive relationship was found for this variable due to the number of workers studied and the need for a longer follow-up period (years) to reveal the harmful impacts of a low AMD on workers’ health. It would be interesting to continue to collect data from a larger sample of workers. Similar studies [37] have found that a diet high in fruit, vegetables and fish but low in meat was associated with shorter periods of sick leave than a diet high in animal fats. Herruzo et al. [28] studied a population of healthcare workers and attempted to identify a relationship between healthy lifestyles and risk factors for heart disease. AMD and PA were two of the variables that they considered in their study. Their results differed from those of our study, as the percentage of workers with high AMD was 68.3% of the total, compared to 59.2% of the workers in our study. They also observed a relationship between higher AMD and levels of PA, with a tendency for workers who engaged in more vigorous PA to have a higher AMD.

The results of this study underline the complexity of PA strategies and engagement in AMD. For future research, longer follow-ups could show changes in workers’ health status with high AMD; and further explore the characteristics of the physical activity undertaken, identifying specific exercises that could lead to health improvements.

The implications of these results turn out to be essential as a starting point to develop effective PA and diet interventions in the workplace. Due to a change in lifestyle patterns, an integrative assessment is needed where the structure of Total Worker Health is implemented [38], taking into consideration individual circumstances, and overcoming bias when integrating this approach into the workplace setting [32,39].

### Study Limitations

This study has a number of limitations. With regard to the sample, it was not possible to include a larger sample size on this occasion but it would be interesting to do so in future research. It is also possible that the workers did not complete the questionnaires honestly or suffered from recall bias. Some bias is inherent to the tests used, especially the IPAQ, which refers to PA in the past week and may not reflect workers’ typical PA habits.

## 5. Conclusions

This study has demonstrated a significant relationship between vigorous PA and a reduction in the duration of TD. The mean duration of TD among workers with a PA of >3000 METs per week was 28.1 days shorter than among workers who were sedentary or who engaged in light-moderate PA. When all other variables (age, gender, tobacco use, alcohol consumption, type of work) were equal, workers engaging in vigorous PA experienced 35 days less TD than workers with a lower intensity of PA.

Type of work was found to be associated with a reduction in days of TD, with administrative professionals losing the fewest days on average, followed by security professionals and trade professionals.

No relationship was found between a high AMD and workers’ health, despite half of the workers displaying good AMD.

Occupational health policies should implement specific strategies to promote healthy lifestyles in response to workers’ needs. These results highlight the need to study in depth those aspects that can influence workers’ health, especially personal characteristics, lifestyles, type of work, and socioeconomic aspects, which can determine the acquisition or not of certain healthy habits.

## Figures and Tables

**Figure 1 ijerph-19-14932-f001:**
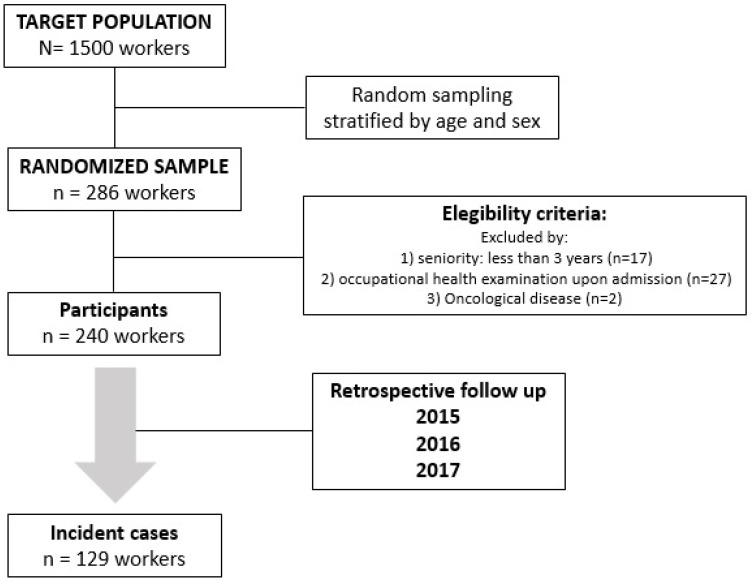
Flowchart of study participants.

**Table 1 ijerph-19-14932-t001:** Characteristics of sample and comparison with reference population.

Variable	Study Sample *n* = 240	Reference Population N = 1500	*p*
Men	177 (73.8%)	1039 (69.3%)	0.18
Women	63 (26.2%)	461 (30.4%)	0.18
Age (men)	50.2 (8.3)	49.5 (7.9)	0.28
Age (women)	50.5 (6.3)	50 (7.2)	0.6
Age (total)	50.2 (7.9)	49.7 (7.7)	0.35
Administrative professions	109 (45.4%)	565 (37.7%)	0.02
Trade professions	59 (24.6%)	419 (27.9%)	0.33
Security professions	72 (30%)	474 (31.6%)	0.79

**Table 2 ijerph-19-14932-t002:** Predictor variables and mean duration of common illness/workplace injury.

Variable	*n* (%)	Duration Mean ± SD	Median Duration (IQR)	Range (Max-min)	*p*
Gender					0.4
Men	177 (73.8%)	49.3 ± 97.8	7 (57.5)	0–741	
Women	63 (26.2%)	37.1 ± 103.8	0 (23)	0–608	
Alcohol consumption					0.08 ^a^
None/Low	192 (80%)	40.4 ± 85.9	2 (45)	0–608	
Moderate/High	48 (20%)	68.7 ± 139.6	11 (75.8)	0–741	
Tobacco use					0.37 ^a^
Non-smoker	123 (51.25%)	38.7 ± 72.8	4 (48)	0–432	
Smoker	44 (18.3%)	72.3 ± 143.6	6 (87.5)	0–741	
Ex-smoker	73 (30.4%)	42.6 ± 104.3	0 (37	0–608	
Type of work					0.001 ^a^
Administrative professions	109 (45.4%)	29.5 ± 75.1	0 (20.5)	0–441	
Trade professions	59 (24.6%)	74.5 ± 149	8 (78)	0–741	
Security professions	72 (30%)	47.8 ± 73.9	15.5 (77.8)	0–417	

^a^ Non-parametric test for comparison of means: Mann-Whitney’s *U*-test/Kruskall-Wallis test; SD: Standard deviation; IQR (interquartile range).

**Table 3 ijerph-19-14932-t003:** Components of AMD and duration of common illness/workplace injury.

Components and Portions		*n* (%)	Duration Mean ± SD	*p*
Olive oil (teaspoons per day)	<2	17 (7.1%)	37.2 ± 57.4	
	≥2	223 (92.9%)	46.7 ± 101.8	0.7
Vegetables (servings per day)	<2	149 (62.1%)	51.1 ± 106.3	
	≥2	91 (37.9%)	37.7 ± 86.6	0.3
Fruit (servings per day)	<3	146 (66.8%)	45.2 ± 104.6	
	≥3	94 (33.2%)	47.4 ± 90.9	0.87
Red meat (servings per day)	<1	102 (42.5%)	32.4 ± 59.9	
	≥1	138 (57.5%)	56.2 ± 119.6	0.067
Butter (servings per day)	<1	214 (89.2%)	45.3 ± 98	
	≥1	26 (10.8%)	51.9 ± 111.4	0.75
Sugary drinks (per day)	<1	183 (76.25%)	47.5 ± 107.6	
	≥1	57 (23.75%)	41.6 ± 66.4	0.7
Wine (glasses per week)	<3	175 (72.9%)	41.6 ± 91.4	
	≥3	64 (27.1%)	59 ± 118.5	0.23
Pulses (servings per week)	<3	141 (58.75%)	44.9 ± 88.4	
	≥3	99 (41.25%)	47.7 ± 113.5	0.8
Fish (servings per week)	<3	162 (67.5%)	45.6 ± 97.7	
	≥3	78 (32.5%)	47 ± 103.2	0.9
Baked goods (servings per week)	<3	188 (78.3%)	49.8 ± 97.7	0.27
	≥3	52 (21.7%)	47.7 ± 113.5	
Nuts (servings per week)	<1	38 (15.8%)	70.3 ± 114.7	
	≥1	202 (84.2%)	41.5 ± 95.7	0.1
Fried food (servings per week)	<2	32 (13.3%)	51 ± 130.1	
	≥2	208 (86.7%)	45.3 ± 94	0.34

SD: Standard Deviation.

**Table 4 ijerph-19-14932-t004:** Level of physical activity and duration of common illness/workplace injury.

Level of Physical Activity		*n* (%)	Duration Mean ± SD	Duration Median (IQR)	Range (Max–Min)	*p*
3 categories	Sedentary–light Moderate High	11 (4.6%)	40.7 ± 59	3 (114)	0–145	0.67 *
		72 (30%)	68 ± 146.4	0 (78.8)	0–741	
		157 (65.4%)	36.3 ± 69.4	4 834.5)	0–417	
2 categories	Sedentary–light–moderate High	83 (34.6%)	64.4 ± 138.1	0 (80)	0–744	0.085
		157 (65.4%)	36.3 ± 69.4	7 (45)	0–444	
Quartiles (METs)	Q1 (<2455)	60 (25%)	84 ± 157.36	1.5 (96.5)	0–741	0.002
	Q2 (2455–4000)	60 (25%)	19.4 ± 39.1	0 (18.5)	0–212	
	Q3 (4001–5700)	60 (25%)	32.2 ± 65.2	3.5 (39)	0–417	
	Q4 (>5700)	60 (25%)	48.6 ± 84	13 (60.3)	0–444	

* Note: Non-parametric test for comparison of means: Mann–Whitney’s *U*-test/Kruskal–Wallis test; SD: Standard deviation, IQR (interquartile range).

**Table 5 ijerph-19-14932-t005:** Multiple linear regression adjusted for gender, tobacco use, alcohol consumption and type of work.

Variable	Coefficient	Standard Error	95% CI	*p*
Trade professions	51.5	15.8	20.2 to 82.7	<0.01
Security professions	24	14.9	−5.3 to 53.3	0.11
High level of PA	−35.4	13.4	−61.7 to −9	<0.01
Constant	49.3	11.9	-	-

Note: CI—Confidence Interval.

## Data Availability

The data presented in this study are available on request from the corresponding author. The data are not publicly available due to company rules.

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
