# Peer review of "Lifestyle, Type of Work, and Temporary Disability: An Incidence Study of the Working Population"

_ijerph, 2022, doi:10.3390/ijerph192214932_

Round 1
Reviewer 1 Report
Dear authors: Your study provides important evidence on the relationship between physical activity and temporary disability among working adults. Prior to publication, this well-written manuscript would benefit from several minor revisions. For your consideration, please see my suggestions below.
In the Introduction, on lines 46 - 50, you write: "For this reason, companies are increasingly opting to organise health promotion activities in the workplace to encourage healthy eating habits and increase adherence to physical activity (PA) among employees with the aim of improving their health and wellbeing, creating a better working environment, and enhancing productivity in the workplace (Tamers, 2020)." In the reference list, you include this information from a presentation: "Tamers, S. (2020) Total worker health® is the future of worker safety, health, and well-being. European Journal of Public Health, 324 30(5), 165-169. https://doi.org/10.1093/eurpub/ckaa165.169." However, it would be more appropriate to cite the full journal article upon which this presentation was based. The more appropriate reference would be: "Tamers, S. L., Streit, J., Pana‐Cryan, R., Ray, T., Syron, L., Flynn, M. A., ... & Howard, J. (2020). Envisioning the future of work to safeguard the safety, health, and well‐being of the workforce: A perspective from the CDC's National Institute for Occupational Safety and Health. American journal of industrial medicine, 63(12), 1065-1084. https://doi.org/10.1002/ajim.23183"
In the Introduction, on lines 71-74, you write: "Nevertheless, these interventions must be adapted to the specific needs of each company, the type of work, and the personal characteristics of employees. To achieve this, workers’ lifestyles and loss of health, as well as their impact on absenteeism, should be analysed before any intervention is carried out." How would you connect this statement from the Introduction to your study's results? In the Discussion and/or Conclusion section(s), please discuss how this study's findings should influence workplace practices and/or future research. What are your study's implications? Do you believe that the findings provide evidence for implementing Total Worker Health principles to allow and encourage more physical activity among workers?
In the Materials and Methods section, when describing the Study Design on Line 84 there is a typo. "Simple size" should be corrected to "sample size"
In the Materials and Methods section, when describing Eligibility Criteria on Lines 87 - 93, you state: "Inclusion criteria. Workers who had attended an occupational health assessment during the study period and were working at the start of the study were included. Exclusion criteria. Workers who had been at their current workplace for less than 3 years, workers who had undergone health assessments upon starting their jobs, and workers who were on sick leave due to cancer during the study period were excluded from the study." What is the difference between "had attended an occupational health assessment during the study period" for the inclusion criteria and "had undergone health assessments upon starting their jobs" for exclusion criteria? These descriptions are confusing as they seem to be the same thing. Please explain in a bit more detail to differentiate these criteria.
In the Materials and Methods section, please provide an explanation of the data collection methods. You conducted a survey or questionnaire? Please include a copy of the data collection instrument as an appendix. What year was the survey conducted? How were study participants recruited? Figure 1 states there was random sampling, but this is not described in the Methods section.
In the Results section, on lines 134-135, you mention a reference population. Who was included in the reference population? Where did you obtain those data? Please explain all these details in the Methods section.
In the Discussion section, on lines 215-218, you state: "However, it is not immediately clear why workers do not engage in vigorous PA. It is possible that workers do not engage in PA or do so with low-moderate intensity because their physical fitness does not permit them to perform vigorous exercise or simply because they do not wish to do so (Proper et al., 2006)." Please consider the environmental, economic, and social factors that might influence workers' ability to engage in vigourous physical activity. As public health researchers, it is important that we highlight the social and structural determinants of health, rather than merely focusing on individuals. It is unfortunate when researchers blame people for their poor health solely due to personal failings, without a consideration of the larger contexts in which we all live. A great strength of this study is that it focuses on the workplace, which is an environmental and structural influence on people's health. People's incomes, their neighborhoods, their transportation, and the amount of free time that they have outside of paid work and family responsibilities can all influence their ability to engage in physical activity. Please see the following references for some background on these issues:
Ball, K., Carver, A., Downing, K., Jackson, M., & O'Rourke, K. (2015). Addressing the social determinants of inequities in physical activity and sedentary behaviours. Health promotion international, 30(suppl_2), ii8-ii19. https://doi.org/10.1093/heapro/dav022
World Health Organization (2004). Global strategy on diet, physical activity and health. https://www.who.int/publications/i/item/9241592222
In this global strategy on physical activity and health, WHO states that, "Multisectoral policies are needed to promote physical activity. National policies to promote physical activity should be framed, targeting change in a number of sectors. Governments should review existing policies to ensure that they are consistent with best practice in population-wide approaches to increasing physical activity. (1) Framing and review of public policies. National and local governments should frame policies and provide incentives to ensure that walking, cycling and other forms of physical activity are accessible and safe; transport policies include nonmotorized modes of transportation; labour and workplace policies encourage physical activity; and sport and recreation facilities embody the concept of sports for all. Public policies and legislation have an impact on opportunities for physical activity, such as those concerning transport, urban planning, education, labour, social inclusion, and health-care funding related to physical activity. Community involvement and enabling environments. Strategies should be geared to changing social norms and improving community understanding and acceptance of the need to integrate physical activity into everyday life. Environments should be promoted that facilitate physical activity, and supportive infrastructure should be set up to increase access to, and use of, suitable facilities."
Author Response
Comments and Suggestions for Authors
Dear authors: Your study provides important evidence on the relationship between physical activity and temporary disability among working adults. Prior to publication, this well-written manuscript would benefit from several minor revisions. For your consideration, please see my suggestions below.
- In the Introduction, on lines 46 - 50, you write: "For this reason, companies are increasingly opting to organise health promotion activities in the workplace to encourage healthy eating habits and increase adherence to physical activity (PA) among employees with the aim of improving their health and wellbeing, creating a better working environment, and enhancing productivity in the workplace (Tamers, 2020)." In the reference list, you include this information from a presentation: "Tamers, S. (2020) Total worker health® is the future of worker safety, health, and well-being. European Journal of Public Health, 324 30(5), 165-169. https://doi.org/10.1093/eurpub/ckaa165.169." However, it would be more appropriate to cite the full journal article upon which this presentation was based. The more appropriate reference would be: "Tamers, S. L., Streit, J., Pana‐Cryan, R., Ray, T., Syron, L., Flynn, M. A., ... & Howard, J. (2020). Envisioning the future of work to safeguard the safety, health, and well‐being of the workforce: A perspective from the CDC's National Institute for Occupational Safety and Health. American journal of industrial medicine, 63(12), 1065-1084. https://doi.org/10.1002/ajim.23183"
Response:
Thank you very much for your comment. The authors have changed the citation in the manuscript.
- In the Introduction, on lines 71-74, you write: "Nevertheless, these interventions must be adapted to the specific needs of each company, the type of work, and the personal characteristics of employees. To achieve this, workers’ lifestyles and loss of health, as well as their impact on absenteeism, should be analysed before any intervention is carried out." How would you connect this statement from the Introduction to your study's results?
Response:
Thank you very much for your comment.
This perspective is included into the Total Workers Health approach and is essential for the implications of the study. We have added in the discussion section this connection wiht the following paragraph:
“The results of this study underline the complexity of PA strategies and engagement in AMD. For future research, longer follow-ups could show changes in workers’ health status with high AMD; and further explore the characteristics of the physical activity undertaken, identifying specific exercises that could lead to health improvements.”
- In the Discussion and/or Conclusion section(s), please discuss how this study's findings should influence workplace practices and/or future research.
Response:
Thank you very much for your comment. The following paragraph has been added to the Conclusions subsection:
“Occupational health policies should implement specific strategies to promote healthy lifestyles in response to workers' needs. These results highlight the need to study in depth those aspects that can influence workers' health, especially personal characteristics, lifestyles, type of work, and socioeconomic aspects, which can determine the acquisition or not of certain healthy habits”.
- What are your study's implications? Do you believe that the findings provide evidence for implementing Total Worker Health principles to allow and encourage more physical activity among workers?
Response:
Thank you very much for this comment.
We have added the study’s implications in the discussion section as long with point 3:
“The implications of these results turn out to be essential as a starting point to develop effective PA and diet interventions in the workplace. Due to a change in lifestyle patterns, an integrative assessment is needed where the structure of Total Worker Health is implemented [38], taking into consideration individual circumstances, and overcoming bias when integrating this approach into the workplace setting [32,39].”
- In the Materials and Methods section, when describing the Study Design on Line 84 there is a typo. "Simple size" should be corrected to "sample size"
Response:
Thank you very much for your appreciation. “Sample size” has been corrected in the manuscript.
- In the Materials and Methods section, when describing Eligibility Criteria on Lines 87 - 93, you state: "Inclusion criteria. Workers who had attended an occupational health assessment during the study period and were working at the start of the study were included. Exclusion criteria. Workers who had been at their current workplace for less than 3 years, workers who had undergone health assessments upon starting their jobs, and workers who were on sick leave due to cancer during the study period were excluded from the study." What is the difference between "had attended an occupational health assessment during the study period" for the inclusion criteria and "had undergone health assessments upon starting their jobs" for exclusion criteria? These descriptions are confusing as they seem to be the same thing. Please explain in a bit more detail to differentiate these criteria.
Response:
Thank you very much for your comment. The second exclusion criterion "workers who had undergone health assessments upon starting their jobs" is already included in the first exclusion criterion "Workers who had been at their current workplace for less than 3 years"; therefore, the second exclusion criterion has been eliminated.
- In the Materials and Methods section, please provide an explanation of the data collection methods. You conducted a survey or questionnaire? Please include a copy of the data collection instrument as an appendix. What year was the survey conducted?
Response:
Thank you very much for your comment.
The authors carried out the study through interviews with the workers during the occupational health checkup carried out in the study period, that is, from 2015 to 2017. In these interviews, questionnaires were used to collect lifestyles as physical activity (IPAQ questionnaire) and adherence to the Mediterranean diet (modified MEDAS questionnaire). In addition, physical examination tests were carried out, personal and work records were updated, etc.
The information collection instruments (questionnaires) are described in Material and Methods as well as in the References section.
- How were study participants recruited?
Response:
Participants were recruited from periodic occupational health checkups conducted through occupational health surveillance. A doctor (CAF) and occupational health nurse (MRS) were in charge of carrying out the check-ups on the workers.
- Figure 1 states there was random sampling, but this is not described in the Methods section.
Response:
Thank you very much for this comment.
Random sampling is too indicated in the Methods section, specifically in 2.1.Study design. Population and sample: “Workers who attended occupational health assessments during the study period were selected at random and stratified by age and sex”. However, for better understanding, this sentence has been moved to the end of the following paragraph: “Participants were selected through random sampling stratified by age and sex”.
- In the Results section, on lines 134-135, you mention a reference population. Who was included in the reference population? Where did you obtain those data? Please explain all these details in the Methods section.
Response:
Thank you very much for your comment.
Table 1 shows the comparison of the study sample (n=240) with the reference population (N= 1,500) from which the sample was extracted, in order to show that there are no significant differences (except in the administrative job group). This shows that the study sample was conveniently drawn and is representative of the reference population.
The reference population was made up of 1,500 workers from the Córdoba City Council. This is indicated in the Material and Methods section.
- In the Discussion section, on lines 215-218, you state: "However, it is not immediately clear why workers do not engage in vigorous PA. It is possible that workers do not engage in PA or do so with low-moderate intensity because their physical fitness does not permit them to perform vigorous exercise or simply because they do not wish to do so (Proper et al., 2006)." Please consider the environmental, economic, and social factors that might influence workers' ability to engage in vigourous physical activity. As public health researchers, it is important that we highlight the social and structural determinants of health, rather than merely focusing on individuals. It is unfortunate when researchers blame people for their poor health solely due to personal failings, without a consideration of the larger contexts in which we all live. A great strength of this study is that it focuses on the workplace, which is an environmental and structural influence on people's health. People's incomes, their neighborhoods, their transportation, and the amount of free time that they have outside of paid work and family responsibilities can all influence their ability to engage in physical activity. Please see the following references for some background on these issues:
Ball, K., Carver, A., Downing, K., Jackson, M., & O'Rourke, K. (2015). Addressing the social determinants of inequities in physical activity and sedentary behaviours. Health promotion international, 30(suppl_2), ii8-ii19. https://doi.org/10.1093/heapro/dav022
World Health Organization (2004). Global strategy on diet, physical activity and health. https://www.who.int/publications/i/item/9241592222
In this global strategy on physical activity and health, WHO states that, "Multisectoral policies are needed to promote physical activity. National policies to promote physical activity should be framed, targeting change in a number of sectors. Governments should review existing policies to ensure that they are consistent with best practice in population-wide approaches to increasing physical activity. (1) Framing and review of public policies. National and local governments should frame policies and provide incentives to ensure that walking, cycling and other forms of physical activity are accessible and safe; transport policies include nonmotorized modes of transportation; labour and workplace policies encourage physical activity; and sport and recreation facilities embody the concept of sports for all. Public policies and legislation have an impact on opportunities for physical activity, such as those concerning transport, urban planning, education, labour, social inclusion, and health-care funding related to physical activity. Community involvement and enabling environments. Strategies should be geared to changing social norms and improving community understanding and acceptance of the need to integrate physical activity into everyday life. Environments should be promoted that facilitate physical activity, and supportive infrastructure should be set up to increase access to, and use of, suitable facilities."
Response:
We are grateful for the citations and are now included as long as other in the discussion section:
“Social determinants of health also have a huge impact on workers’ PA opportunities. Some authors highlight that PA intervention strategies may target/address social aspects such as economic level, environment, and access to health and social services [30, 31]. Therefore, it is essential to consider those aspects and study vulnerable workers’ needs, designing safe and healthy spaces in the workplace [32], as well as strong and supportive policies that help overcome those barriers [33, 34].”
- Ball, K.; Carver, A.; Downing, K.; Jackson, M.; O'Rourke, K. Addressing the social determinants of inequities in physical activity and sedentary behaviours. Health Promot. Int. 2005, 30, ii8-ii19; https://doi.org/10.1093/heapro/dav022
- Laddu, D.; Paluch, A. E.; LaMonte, M. J. The role of the built environment in promoting movement and physical activity across the lifespan: Implications for public health. Prog Cardiovasc Dis. 2021, 64, 33–40; https://doi.org/10.1016/j.pcad.2020.12.009
- Lobczowska, K.; Banik, A.; Forberger, S.; Kaczmarek, K.; Kubiak, T.; Neumann-Podczaska, A.; Romaniuk, P.; Scheidmeir, M.; Scheller, D. A.; Steinacker, J. M.; Wendt, J.; Bekker, M.; Zeeb, H.; Luszczynska, A. Policy Evaluation Network (PEN) Consortium. Social, economic, political, and geographical context that counts: meta-review of implementation determinants for policies promoting healthy diet and physical activity. BMC Public Health 2022, 22, 1055; https://doi.org/10.1186/s12889-022-13340-4
- World Health Organization. Global strategy on diet, physical activity and health. Available online: https://www.who.int/publications/i/item/9241592222 (accessed on 02 November 2022)
- Bantham, A.; Taverno Ross, S. E.; Sebastião, E.; Hall, G. Overcoming barriers to physical activity in underserved populations. Prog Cardiovasc Dis. 2021, 64, 64–71; https://doi.org/10.1016/j.pcad.2020.11.002

Reviewer 2 Report
Authors affiliations
- Affiliations number 1, 4, 5 and 7 are the same. So this information must be reviewed, in authors superscript and in affiliations description.
- In authors 1, 3 and 4, the institutional email is recommended.
Abstract
- In line 33, number and percentages of men and women are not correct in comparison with line 26 and lines 132-133.
Keywords
Healthy diet is dispensable; health promotion and mediterranean diet can be maintained in its place.
Introduction
- An in-depth look in this section is recommended, maybe through bibliography extension (2021 and 2022 references).
- There is not a research hypothesis.
Materials and methods
- Data obtained from 2015 to 2017. Why not until 2021 or 2022?
- Place name could be mentioned (line 81); organizations and entities name do not appear along the manuscript, so anonymization is preserved.
- 'Study design' and 'Population and sample' must be in different sections.
- Recruitment procedure is not described.
- Age, sex and kind of work are not mentioned in sample section.
- Variables are not sufficiently described or defined.
- Protocols and methods choice are not justified. Why these are the most suitable for this study?
- In 'Ethical and legal aspects', the review board name must be mentioned. In fact, it appears in 'Etical aspects' after Conclusions.
Results
- Men and women % (lines 132-133) are different in relation to Abstract.
- Tables description is incomplete. Abbreviations are not defined, and there is not value for significant difference (for example: * if p<0.05, or ** if p<0,01).
- Data in tables are not correctly expressed in Tables 2, 3 and 4. Data should be expressed as mean +/- SD. Mean (SD) is not appropriated.
- In Tables 2-4, SD value is higher than mean value in all items. Is this OK? Significant differences reliability must be reviewed.
- p value must be reviewed along the full text and tables, because unit is sometimes missed. For example, in line 179, "p<.05", but this should be "p<0.05". An exhaustive correction in the whole manuscript is required.
- In tables where there is p<0.05 (administrative professions in Table 1, 2 categories in Table 4), p value should be exact, and the significance be marked with a symbol (for example, "*", "a", etc.).
Discussion
- There are not references from 2020 to 2022. A bibliography updating is required.
- Discussion content is based on results comparison to other studies. Nevertheless, an explanation or reasoning for all results is missed.
Conclusions
Finally, the most interesting relationship was between physical activity and temporary disability. This study should have been more focused in this variable, distinguising type of exercise/activity, intensity, volume, frequency, etc. Level of physical activity (sedentary-light, moderate, high) or METs quartiles are not enough for exercise categorization.
Author Response
POINT BY POINT ANSWER DOCUMENT
REVIEWER 2
- Authors affiliations
- Affiliations number 1, 4, 5 and 7 are the same. So this information must be reviewed, in authors superscript and in affiliations description. - In authors 1, 3 and 4, the institutional email is recommended.
Response:
Thank you very much for your comment.
Authors who share work affiliations have been unified in the same superscript. In addition, personal emails have been replaced by institutional emails. The modifications have been carried out in the manuscript (red colour).
- Abstract
- In line 33, number and percentages of men and women are not correct in comparison with line 26 and lines 132-133.
Response:
The percentages on line 33 are calculated based on the total number of men (104/177= 58.8%) and the total number of women (25/63=39.7%). However, the percentages of lines 132-133 are calculated over the entire sample, that is, for men (177/240= 73.8%) and for women (63/240= 26.2%).
- Keywords
Healthy diet is dispensable; health promotion and mediterranean diet can be maintained in its place.
Response:
Thank you very much for this comment. The authors agree with this appreciation.
“Healthy diet” has been removed.
- Introduction
- An in-depth look in this section is recommended, maybe through bibliography extension (2021 and 2022 references).
Response:
Thank you very much for your comment. The references have been updated in the introduction section, and the following have been added:
- Yammine, A.; Namsi, A.; Vervandier-Fasseur, D.; Mackrill, J.J.; Lizard, G.; Latruffe, N. Polyphenols of the Mediterranean diet and their metabolites in the prevention of colorectal cancer. Molecules 2021, 26, 3483; https://doi.org/10.3390/molecules26123483
- Farràs, M.; Almanza-Aguilera, E.; Hernáez, Á.; Agustí, N.; Julve, J.; Fitó, M.; Castañer, O. Beneficial effects of olive oil and Mediterranean diet on cancer physio-pathology and incidence. Semin. Cancer Biol. 2021, 73, 178-195; https://doi.org/10.1016/j.semcancer.2020.11.011
- Albulescu, P.; Macsinga, I.; Rusu, A.; Sulea, C.; Bodnaru, A.; Tulbure, B. T. "Give me a break!" A systematic review and meta-analysis on the efficacy of micro-breaks for increasing well-being and performance. PLoS One 2022, 17, e0272460; https://doi.org/10.1371/journal.pone.0272460
- Pronk N. P. Implementing movement at the workplace: Approaches to increase physical activity and reduce sedentary behavior in the context of work. Prog Cardiovasc Dis. 2021, 64, 17–21; https://doi.org/10.1016/j.pcad.2020.10.004
- There is not a research hypothesis.
Response:
Thank you very much for your comment.
The research hypothesis has been added in the Introduction section.
“The research hypothesis was to find out if those workers who performed moderate-high physical activity or high adherence to the Mediterranean diet presented less loss of health due to temporary disability”.
- Materials and methods
- Data obtained from 2015 to 2017. Why not until 2021 or 2022?
Response:
Thank you very much for this comment.
The study period was limited to the doctoral stay of the leading researcher (María del Rocío Jiménez Mérida), which ended in 2017. We regret that we do not have additional data after 2017.
- Place name could be mentioned (line 81); organizations and entities name do not appear along the manuscript, so anonymization is preserved.
Response:
The name of the institution where the study was carried out (Cordoba City Council - Spain) is now indicated in the manuscript.
- 'Study design' and 'Population and sample' must be in different sections.
Response:
Thank you very much for your comment.
The “Population and sample” subsection has now been created in the manuscript.
- Recruitment procedure is not described.
Response:
Thank you very much for your comment.
The authors have added a new sentence in the “Eligibility criteria” subsection: “Participants were recruited from regular occupational health checkups that they carried out in the company".
- Age, sex and kind of work are not mentioned in sample section.
Response:
Thank you very much for your comment.
The variables age, sex and type of work are described in the subsection “Study and measurement variables”. The authors consider that the information on these variables is complete. However, we are available to expand this information if the reviewer considers it appropriate.
- Variables are not sufficiently described or defined.
Response:
Thank you very much for your comment.
The authors have revised the description of the study variables and the information has been expanded.
- Protocols and methods choice are not justified. Why these are the most suitable for this study?
Response:
Thank you very much for your comment.
The following protocols have been selected:
- Alcohol consumption: WHO guidelines, widely accepted by most countries, have been used.
- Adherence to the Mediterranean diet: the Trichopoulou questionnaire was used as it is a validated and reliable tool for the Spanish population.
- Physical activity: the IPAQ questionnaire has been used as it is a reliable and valid tool to quantify and categorize the physical activity carried out by the worker, both during their working day and leisure time.
- In 'Ethical and legal aspects', the review board name must be mentioned. In fact, it appears in 'Etical aspects' after Conclusions.
Response:
Thank you very much for your comment.
The review board name has been included in the “Ethical and legal aspects” subsection.
- Results
- Men and women % (lines 132-133) are different in relation to Abstract.
Response:
- Tables description is incomplete. Abbreviations are not defined, and The percentages on line 33 are calculated based on the total number of men (104/177= 58.8%) and the total number of women (25/63=39.7%). However, the percentages of lines 132-133 are calculated over the entire sample, that is, for men (177/240= 73.8%) and women (63/240= 26.2%).
there is not value for significant difference (for example: * if p<0.05, or ** if p<0,01).
Response:
Thank you very much for your comment.
The tables have been revised and all abbreviations have been added.
- Data in tables are not correctly expressed in Tables 2, 3 and 4. Data should be expressed as mean +/- SD. Mean (SD) is not appropriated.
Response:
Thank you very much for your comment.
Tables 2, 3 and 4 have been revised and mean +/- SD has been included.
- In Tables 2-4, SD value is higher than mean value in all items. Is this OK? Significant differences reliability must be reviewed.
Response:
Thank you very much for your comment.
The outcome variable in our study was the loss of health of the workers due to temporary disability (measured in number of days). This variable has shown a range of 0 to 741 days, with a mean of 46.1 days and a standard deviation of 99.3 days. Due to this high dispersion of the data, the median and interquartile range have been included as the most appropriate statistics to represent the duration of the health loss processes.
This high heteroscedasticity has caused the data not to fit a normal distribution, and therefore, non-parametric tests (Kruskal-Wallis or Mann-Whitney U) were used for the comparison of means.
The authors have included the median and interquartile range in Tables 2 and 4.
- p value must be reviewed along the full text and tables, because unit is sometimes missed. For example, in line 179, "p<.05", but this should be "p<0.05". An exhaustive correction in the whole manuscript is required.
Response:
Thank you very much for your comment.
The Results section has been revised and all units of p-value have been indicated
- In tables where there is p<0.05 (administrative professions in Table 1, 2 categories in Table 4), p value should be exact, and the significance be marked with a symbol (for example, "*", "a", etc.).
Response:
Thank you very much for your comment.
All p-values in tables have been checked and exact values shown.
- Discussion
- There are not references from 2020 to 2022. A bibliography updating is required.
Response:
Thank you very much for your comment. The references have been updated in the discussion section, and the following have been added:
- Laddu, D.; Paluch, A. E.; LaMonte, M. J. The role of the built environment in promoting movement and physical activity across the lifespan: Implications for public health. Prog Cardiovasc Dis. 2021, 64, 33–40; https://doi.org/10.1016/j.pcad.2020.12.009
- Lobczowska, K.; Banik, A.; Forberger, S.; Kaczmarek, K.; Kubiak, T.; Neumann-Podczaska, A.; Romaniuk, P.; Scheidmeir, M.; Scheller, D. A.; Steinacker, J. M.; Wendt, J.; Bekker, M.; Zeeb, H.; Luszczynska, A. Policy Evaluation Network (PEN) Consortium. Social, economic, political, and geographical context that counts: meta-review of implementation determinants for policies promoting healthy diet and physical activity. BMC Public Health 2022, 22, 1055; https://doi.org/10.1186/s12889-022-13340-4
- World Health Organization. Global strategy on diet, physical activity and health. Available online: https://www.who.int/publications/i/item/9241592222 (accessed on 02 November 2022)
- Bantham, A.; Taverno Ross, S. E.; Sebastião, E.; Hall, G. Overcoming barriers to physical activity in underserved populations. Prog Cardiovasc Dis. 2021, 64, 64–71; https://doi.org/10.1016/j.pcad.2020.11.002
- Lemke M. K. Is the Total Worker Health Program Missing Its Mark?: Integrating Complex Systems Approaches to Unify Vision and Epistemology. J Occup Environ Med. 2021, 63(5), e304–e307; https://doi.org/10.1097/JOM.0000000000002183
- Sorensen, G.; Dennerlein, J. T.; Peters, S. E.; Sabbath, E. L.; Kelly, E. L.; Wagner, G. R. The future of research on work, safety, health and wellbeing: A guiding conceptual framework. Soc Sci Med. 2021, 269, 113593; https://doi.org/10.1016/j.socscimed.2020.113593
- Discussion content is based on results comparison to other studies. Nevertheless, an explanation or reasoning for all results is missed.
Response:
Thank you very much for your comment. The authors have included the following paragraph in the Discussion section:
“The results of this study underline the complexity of PA strategies and engagement in AMD. For future research, longer follow-ups could show changes in workers’ health status with high AMD; and further explore the characteristics of the physical activity undertaken, identifying specific exercises that could lead to health improvements”.
- Conclusions
Finally, the most interesting relationship was between physical activity and temporary disability. This study should have been more focused in this variable, distinguising type of exercise/activity, intensity, volume, frequency, etc. Level of physical activity (sedentary-light, moderate, high) or METs quartiles are not enough for exercise categorization.
Response:
This information is unavailable from the data collected in the present study, but as highlighted in the conclusion section, this would lead to future research findings.

Round 2
Reviewer 2 Report
In this second version, the authors have implemented the corrections indicated in the previous evaluation.
Please, as a last resort, check every p value along the text (abtract, inclusive), so that 0 unit before '.' is visible.
Author Response
Thank you very much for your comments that have helped significantly to improve the article.
In this manusript revised version, the authors have added all "0" before the decimal point in p-values.